# Biomechanical Properties and Cellular Responses in Pulmonary Fibrosis

**DOI:** 10.3390/bioengineering11080747

**Published:** 2024-07-24

**Authors:** Andong He, Lizhe He, Tianwei Chen, Xuejin Li, Chao Cao

**Affiliations:** 1Department of Engineering Mechanics, Zhejiang University, Hangzhou 310028, China; 2Department of Respiratory and Critical Care Medicine, Key Laboratory of Respiratory Disease of Ningbo, The First Affiliated Hospital of Ningbo University, 59 Liuting Road, Ningbo 315010, China; 3Center for Medical and Engineering Innovation, Central Laboratory, The First Affiliated Hospital of Ningbo University, Ningbo 315010, China; 4Key Laboratory of 3D Printing Process and Equipment of Zhejiang Province, School of Mechanical Engineering, Zhejiang University, Hangzhou 310028, China; 5Key Laboratory of Respiratory Disease of Zhejiang Province, Department of Respiratory and Critical Care Medicine, Second Affiliated Hospital of Zhejiang University School of Medicine, Hangzhou 310009, China

**Keywords:** pulmonary fibrosis, extracellular matrix (ECM), mechanical properties, mechanical stimuli, 3D bioprinting

## Abstract

Pulmonary fibrosis is a fatal lung disease affecting approximately 5 million people worldwide, with a 5-year survival rate of less than 50%. Currently, the only available treatments are palliative care and lung transplantation, as there is no curative drug for this condition. The disease involves the excessive synthesis of the extracellular matrix (ECM) due to alveolar epithelial cell damage, leading to scarring and stiffening of the lung tissue and ultimately causing respiratory failure. Although multiple factors contribute to the disease, the exact causes remain unclear. The mechanical properties of lung tissue, including elasticity, viscoelasticity, and surface tension, are not only affected by fibrosis but also contribute to its progression. This paper reviews the alteration in these mechanical properties as pulmonary fibrosis progresses and how cells in the lung, including alveolar epithelial cells, fibroblasts, and macrophages, respond to these changes, contributing to disease exacerbation. Furthermore, it highlights the importance of developing advanced in vitro models, based on hydrogels and 3D bioprinting, which can accurately replicate the mechanical and structural properties of fibrotic lungs and are conducive to studying the effects of mechanical stimuli on cellular responses. This review aims to summarize the current understanding of the interaction between the progression of pulmonary fibrosis and the alterations in mechanical properties, which could aid in the development of novel therapeutic strategies for the disease.

## 1. Introduction

Pulmonary fibrosis is a progressive and fatal interstitial fibrotic lung disease characterized by fibrosis of the lung tissue. With an overall 5-year survival rate of less than 50%, it is among the most severe non-cancerous lung diseases in terms of prognosis. Due to its aggressive nature and poor outcomes, it is also referred to as a pseudo-tumor disease [1]. According to epidemiological research, approximately 5 million individuals suffer from this disease globally, imposing a significant burden on society [2]. Currently, pulmonary fibrosis remains an irreversible condition with no curative therapies available. As a result, palliative care and lung transplantation are the primary options for improving patient survival [3]. During the progression of pulmonary fibrosis, damage to the alveolar epithelial cells drives the excessive activation of lung fibroblasts. This results in the synthesis of large amounts of the extracellular matrix, leading to the thickening and reconstruction of the lung interstitium [4] (Figure 1A). Consequently, this process deteriorates patients’ pulmonary function, ultimately culminating in respiratory failure. While studies have revealed numerous reasons contributing to pulmonary fibrosis (such as genetic diseases [5], autoimmune diseases [6], occupational exposures [7], toxins [8], and chronic airway diseases [9]), the exact etiology and underlying mechanisms remain elusive.

A key feature of pulmonary fibrosis is the damage to alveoli, the air sacs at the terminal ends of bronchioles. This damage triggers the wound healing process, during which the cells responsible for extracellular matrix (ECM) synthesis (e.g., fibroblast) proliferate actively and contribute to the synthesis, deposition, and remodeling of the ECM, which is an elastic structure comprising collagen, elastin, glycoproteins, and proteoglycans that are cross-linked (Figure 1B). While these cells are rapidly eliminated through apoptosis at the end of the wound healing process [10], they undergo an abnormal response of becoming anti-apoptotic in the pathological process of pulmonary fibrosis, thereby leading to the excessive formation and maturation of the ECM. As a mechanically stiff and elastic substrate for cell adhesion, the ECM provides the mechanical stability and elasticity that are essential in respiratory movements [11]. However, excessive ECM formation results in the formation of scar tissue, which is thicker and stiffer compared to healthy tissue, manifesting in difficulty in breathing [12]. While the changes in mechanical properties of the lung have long been recognized as a consequence of pulmonary fibrosis, increasing evidence suggests that the matrix stiffness, as a mechanical stimulus sensed by cells, regulates the cellular responses, including proliferation and myofibroblast activation [13]. These findings imply that the changes in mechanical properties, previously considered merely outcomes of pulmonary fibrosis, might in return serve as stimuli and modulate the cellular responses that eventually exacerbate the progression of the disease [14]. Based on this concept, an increasing number of studies have been published in recent years highlighting changes in various mechanical properties, including elasticity, viscoelasticity, and surface tension at the surface of alveoli among animal models and patients with pulmonary fibrosis [15,16,17]. These works have reported that altered mechanical properties not only facilitate the onset and progression of pulmonary fibrosis to a certain extent but also hint at the potential novel therapeutic strategies for this deadly disease. To the best of our knowledge, however, there is no comprehensive literature review dedicated to the research of the mechanical stimuli involved in fibrotic lungs and how these works might enlighten the discovery of novel biological targets and corresponding therapeutic strategies.

This narrative review aims to explore the relationship between mechanical stimuli and pulmonary fibrosis. The initial section introduces various mechanical properties in lung tissues that undergo alterations as pulmonary fibrosis progresses. Subsequently, this review focuses on the different types of cells exposed to these mechanical stimuli, examining how their responses are modulated and how the understanding can lead to the identification of biological targets and corresponding medicines/therapeutic strategies for the disease. Recognizing the growing need for models that allow for the precise control of mechanical stimuli to cells for further mechanistic studies, the final section introduces how the development of in vitro cell culture models can meet this demand and what future research is expected in this area. This work endeavors to furnish a concise overview of the biomechanical facets of pulmonary fibrosis while also aiming to spark heightened research interest in this domain.

**Figure 1 bioengineering-11-00747-f001:**
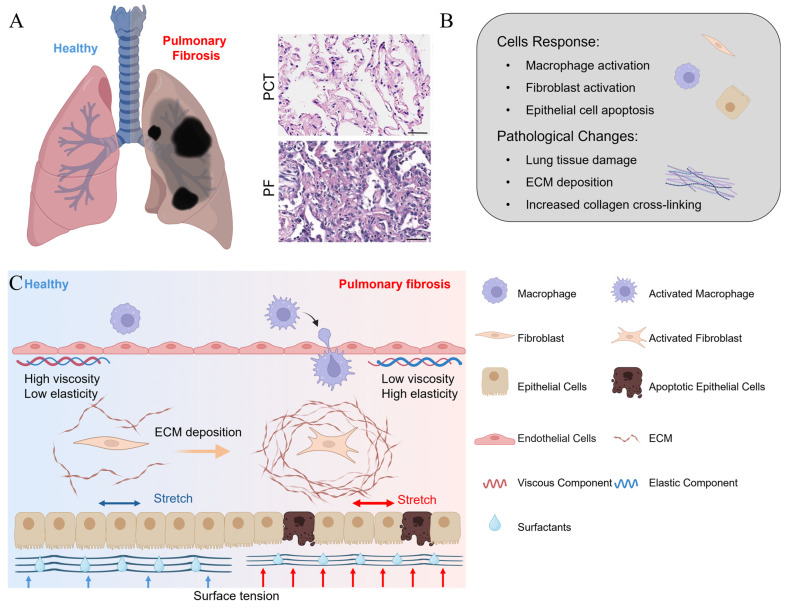
(**A**) A schematic view of a healthy and fibrotic lung [4]; (**B**) the major cellular response and pathological activities as pulmonary fibrosis develops; and (**C**) a schematic representation of changes in the mechanical microenvironment and cell phenotype before and after the onset of pulmonary fibrosis. The elasticity, viscosity, and surface tension contribute to maintaining the homeostasis of healthy lung tissue. As pulmonary fibrosis develops, the elastic and viscous properties of the tissue become altered, along with elevated shear stress and surface tension, which in turn induce the apoptosis of epithelial cells and the activation of immune cells and fibroblasts. The activated fibroblasts then synthesize an excessive amount of the ECM, thereby aggravating the pulmonary fibrosis [14].

## 2. Mechanical Properties of Fibrotic Lung

The progression of pulmonary fibrosis is closely associated with changes in the mechanical stimuli that are applied to and sensed by the cells. The complexity is heightened by the lung’s role in gas exchange [18], as these mechanical changes encompass not only the mechanical properties of tissue and organs but also stimuli originating from the external environment (Figure 1C). It is important to comprehensively understand the changes in mechanical stimuli as both the consequences and driving factors of fibrosis progression [14].

In this section, we discuss the various types of mechanical stimuli involved in pulmonary fibrosis. Many of the reviewed works herein are grounded in experimental methodologies, where these stimuli are quantitatively characterized. Their impacts on the progression of pulmonary fibrosis and the underlying mechanisms are studied via biological assays [14,19]. However, many of the mechanical stimuli are difficult to be experimentally characterized and recapitulated in in vitro/in vivo studies. Consequently, numerical simulations serve as a valuable alternative to experiments. Therefore, the recent advances in the modeling of these stimuli are also briefly discussed [20]. 

### 2.1. Elastic Properties of Fibrotic Lung

The elastic properties of materials dictate their ability to change shape and the tendency to return to their original condition by restoring the bonds between atoms or molecules [21]. In the case of lung tissue, elasticity plays a decisive role in maintaining its structural integrity, primarily attributed to the abundant elastin fibers and collagen fibers. These structural components determine the resilience, deformability, and structural stability of the lung, allowing it to maintain a consistent structural form throughout breathing cycles [22,23].

A characteristic pathological feature of pulmonary fibrosis is the excessive formation of fibers in lung tissue, which results in increased overall stiffness of the lung [24]. Froese investigated the general stiffness of healthy and fibrotic lungs (induced with TGF-β1) on a tissue level. With a constant 5 mN tensile load applied slowly to stretch the lung strips, they noticed a 3-fold increment of Young’s modulus in the fibrotic lung [25]. Microscopically, the Young’s modulus of lung tissue was studied by the nanoindentation method using atomic force microscopy (AFM). As reported by Booth et al., the Young’s modulus of a healthy lung is reported to be 1.96 ± 0.13 kPa, whereas fibrotic lung exhibits a dramatically higher modulus of 16.52 ± 2.25 kPa [26]. With decellularization performed, Young’s modulus for the ECM of the healthy lung was 1.60 ± 0.13 kPa, being insignificantly different from the lung, whereas the ECM for the fibrotic lung was 7.34 ± 0.60 kPa, with the significant reduction in stiffness possibly attributed to the relaxation of matrix proteins. The same trend was also reported by Brown et al., who reported an increment of stiffness from 1.96 ± 1.21 kPa to 17.25 ± 11.06 kPa after fibrosis induction with bleomycin. They further noticed an increasing stiffness of individual RLE-6TN cells (rat alveolar epithelial cell line) from ~1 kPa to ~6 kPa as the hydrogel substrate stiffness increased from 2 to 32 kPa. Combining the findings that a stiffer substrate promoted the expression of α-SMA and TGF-β (both are markers of fibrosis), the authors provided evidence that both the ECM and epithelial cells contributed to the higher stiffness of fibrotic lungs [27].

A much higher standard deviation suggests a heterogeneity in stiffness within fibrotic lungs. This was first examined by Liu et al. using AFM, as they noticed a much higher (~15 kPa) localized stiffness in collagen-rich regions of fibrotic lungs, in addition to regions with stiffness comparable to normal lung tissues [28] (Figure 2A). Another study led by Wu et al. characterized the distinctive mechanical properties within different positions of fibrotic lungs [29]. The key finding was that following 21 days post-lung lobe resection (PNX, a treatment known to exacerbate pulmonary fibrosis in Cdc42-null mice), the average Young’s modulus of the central regions in a fibrotic lung was ~3 kPa, while the fibrotic regions located within a 2 mm region of the lung periphery exhibited a Young’s modulus three times higher (~9 kPa), suggesting that this heterogeneity might drive the progression of pulmonary fibrosis from the periphery of lobes to the central regions (Figure 2B). Additionally, it has been reported that the stiffer ECM in fibrotic lungs effectively induces the differentiation of pulmonary fibroblasts into myofibroblasts, thereby exacerbating the progression of pulmonary fibrosis. This finding suggests that reducing ECM stiffness could be a potential therapeutic strategy for treating pulmonary fibrosis [24]. 

The heterogeneity in the elastic properties of fibrotic lung tissue results in the complex load–displacement behavior of the lung, which can be valuable in the diagnosis, staging, and, potentially, therapy of fibrosis [30,31]. Therefore, researchers are turning to numerical models to precisely recapitulate the mechanical properties of the lung in digital models. In a work by Naini et al., the Ogden, Yeoh, and Polynomial models were employed to simulate the hyperelastic properties of the lung. All three models yielded a good fit to the stress–strain curve of lung tissue, with the average final fitting errors ranging from 2.3% to 6.2%. While the Ogden model produced the most accurate results, a better balance of accuracy and computational efficiency was obtained using the Yeoh model. The authors stated that the models employed are reliable and support accurate biomechanical modeling of lung tissue behavior for various medical applications, such as surgical planning [32]. Classic hyperelastic models, such as the Neo-Hookean model and the Mooney–Rivlin model, have also proven useful in the numerical modeling of lungs [33,34]. For instance, Lian et al. applied the Neo-Hookean model to estimate the ratio and cross-linking degree of collagen fibers within the lungs [35]. Their findings ultimately reveal a close alignment between the estimated and observed conditions in the lung tissue, providing compelling evidence of the inherent elastic material properties of the lung.

### 2.2. Viscoelastic Properties of Fibrotic Lungs

It is known that viscoelasticity is a property that encompasses both the viscosity and elasticity of the viscoelastic material. It describes how the viscoelastic material responds over time to a given mechanical stimulation [36]. The viscosity of the lung was first documented by Bayliss and Robertson in 1939. They developed a model where a pump was supplied with either air or an 80/20 mixture of hydrogen and oxygen and connected to the cat lungs. The total pressure against the stroke with simple harmonic motion was measured, with various frequencies for the harmonic motion applied, and the gas viscosity was controlled by switching to different gases. The authors reported that the lung exhibits viscoelastic properties similar to muscles, and the viscous forces arise from the deformation of lung structures (~15%) and airflow in the air passages (~5%), while the rest (~80%) of the total pressure was attributed to elastic properties [37].

With the source of viscous forces identified, it is reasonable to assume that the viscoelastic properties of the lung change as fibrosis progresses. In 1967, Bachofen and Scherrers investigated lung tissue resistance in 10 patients with pulmonary fibrosis [38]. With the relationship between lung volume (determined by spirometry)–pulmonary resistance (derived from intraesophageal pressure and the airflow rate) depicted in Figure 3A, they noted a dramatic increment in work against viscous lung tissue resistance (shaded area) in patients with pulmonary fibrosis compared to healthy volunteers. While the change was mainly attributed to reduced lung tissue compliance, the authors believed that pathological tissue viscosity was also a reason behind it. In a subsequent study, lung tissues from rats with pulmonary fibrosis induced by bleomycin were analyzed for their viscoelastic properties. The characterization was performed by applying cyclic tension with variable frequency (f = 0.3–10 Hz) and strain (ε = 1–10%), with the total resistance (R) and elastance (E) fitted from the measured tension and length changes. It was reported that compared to the parenchymal tissue of healthy lungs, the overall resistance and elastance remained higher in fibrotic lungs regardless of the frequency and strain. Interestingly, the authors noted that the hysteresivity (η = 2πf (R/E), a dimensionless index reflecting the ratio between the imaginary, out-of-phase stress and the real, elastic stress) was lower in fibrotic lung parenchymal tissues. They attributed the change to the increased biglycan content in fibrotic lungs, assuming that the biglycan lubricated the ECM fibers to reduce the energy dissipation when ECM fibers slide against each other [39,40]. In contrast, Pinart et al. observed that as fibrosis progressed following bleomycin induction, the hysteresivity increased significantly. This increase correlated closely with elevated levels of myeloperoxidase and elastic fibers in the lung tissue. Myeloperoxidase served as a marker for inflammation and neutrophil infiltration, while the elastic fibers were thought to enhance energy dissipation due to increased friction among extracellular matrix (ECM) fibers [41]. Additionally, the excessive formation of fibrosis can lead to an increase in the viscosity of the ECM liquid phase. Under normal conditions, the viscosity is approximately 0.8 cp, whereas under pathological conditions, it can be higher by 1–2 orders of magnitude [42]. Further research has shown that the viscoelastic characteristics of fibrotic tissue are closely related to the degree of collagen cross-linking [43]. Following treatment with the lysyl oxidase (LoX), an enzyme that catalyzes collagen cross-linking, the viscoelastic relaxation time of the collagen networks increased significantly from 7 min to 11 min. Correspondingly, the migration speed of cells cultured on these treated collagen networks also increased (0.5 μm/min vs. 1 μm/min). Based on the acquired images, the mean square displacement (MSD) curves of the cells were calculated, and the cell migration persistence was determined by fitting the data for the first 300 min. The results showed that the cell migration persistence coefficient increased significantly (1.7 vs. 1.8) following LoX treatment. The results demonstrate that regulating the degree of collagen cross-linking can alter the viscoelastic properties of the material and subsequently modulate the cellular responses [44] (Figure 3B).

Recent progress in experimental techniques has facilitated the assessment of the viscoelastic characteristics of biological tissues and hydrogels. Diverse analytical methods are available for the evaluation of viscoelastic test data, including creep and recovery testing, atomic force microscopy (AFM), dynamic mechanical analysis (DMA), nanoindentation, and single-particle-tracking measurements [45,46,47]. By employing these analytical methodologies, researchers can attain a comprehensive understanding of the viscoelastic attributes demonstrated by biological tissues and hydrogels. For example, Tosini et al. utilized nanoindentation to test and characterize the viscoelastic properties of lung tissues and hydrogels at the cell-scale level [48]. They examined samples of pig and human lung tissues alongside a specific GelMA hydrogel to compare their viscoelastic behaviors. Specifically, they analyzed stress–relaxation curves for each sample using a fitting procedure based on a genetic algorithm to assess the relaxation moduli and viscosity parameters. Their results revealed that all samples exhibit load relaxation at a specific indentation depth due to their intrinsic viscoelastic characteristics. Additionally, they found that the GelMA samples have the highest instantaneous shear moduli (~4339 Pa) and the most significant reduction in the relaxation modulus (~1774 Pa), with the decrease being less pronounced in the pig and human lung samples (227~374 Pa). Regarding viscosity properties, the viscosity values for GelMA, pig, and human lung samples are around 848 Pa s, 250 Pa s, and 148 Pa s, respectively. The notably higher viscosity values for the GelMA sample imply a more solid-like behavior compared to the pig and human lung samples. The potential of this method is of considerable value for mechanotransduction research.

More recently, the difference in creep responses of organs that were healthy, fibrotic, or treated with mesenchymal cell infusion was studied by Chang et al., using a murine liver model and an AFM-based microrheology method [49]. The creeping responses of different tissues were characterized with the AFM cantilever attached with a spherical probe, yielding the creep compliance (*J*(*t*), a ratio of strain to stress)–time curves, which were further fitted by two-staged power-law fitting (*J*(*t*)~*t^α^*) over short (*t* = 0.01–0.1 s) and long (*t* = 1–10 s) time scales. The authors reported a reduced power-law exponent as time passes, suggesting the transition of tissue from a highly fluid-like state to a stiffer, elasticity-dominant state. Specifically, the lower power-law exponents for fibrotic tissue remained lower and exhibited a dual-modal distribution, compared to the greater power-law exponents with single-modal distributions in healthy and MSC-treated tissues. Moreover, the compliance was decomposed into tissue–cellular–cytoplasm hierarchies using the self-similar models (Figure 3C). The authors reported that the elasticity of fibrotic tissues remained higher at all levels investigated. In addition, the cytoplasm viscosity showed a heterogeneous, dual-modal distribution, with a significantly higher mean value (59.00 ± 41.52 Pa·s vs. 44.65 ± 9.08 Pa·s) and coefficient of variation (70.38% vs. 20.33%) compared to that of healthy tissue with a single-modal distribution. The treatment of MSC transplantation following fibrosis restored the mode of both the elasticity at all levels and cytoplasm viscosity. Following a receiver operating characteristic (ROC) analysis, it was concluded that the elasticity, especially at the tissue level, was a more effective index than the cytoplasm viscosity in distinguishing fibrotic tissue from healthy tissue (Figure 3C,D). The authors proposed that the change in viscoelastic properties of livers in different conditions could be utilized to quantitatively assess fibrosis. They also indicated the possibility of applying the assessment to other organs as an alternative diagnostic approach.

The investigation of trajectory-specific diffusion of biological tissues and hydrogels has attracted significant attention from both theoretical and experimental communities [50,51,52,53]. For example, Cherstvy et al. studied the passive, thermally driven motion of micron-sized tracers in hydrogels of mucins [50]. They employed the Bayesian approach in conjunction with the nested-sampling algorithm [52,53] to examine tracer diffusion in mucin hydrogels. Their analysis unveiled that tracer diffusion in these gels deviates significantly from Gaussian and ergodic behavior under acidic conditions, indicating the presence of heterogeneous networks. Furthermore, they evaluated the dispersion of individual trajectory time-averaged mean-squared displacement to characterize tracer dynamics in mucin gels. Their findings suggest that the spread of tracers in these gels at the individual trajectory level is mainly governed by Brownian motion (BM) and Fractional Brownian motion (FBM), with occasional occurrences of the diffusing diffusivity (DD) model. Specifically, at pH 7, the BM model seems to prevail in analyzing the single trajectory data. Meanwhile, at pH 4, they noted that the FBM dominates the results predicted by the model. However, at pH 2, they found a more complex scenario with significant variability in hydrogel structure sizes hindering tracer diffusion. This variability stemmed from batch-to-batch discrepancies in the purified mucin, creating a heterogeneous environment. Additionally, they elucidated the distribution of time-averaged displacements and correlations of scaling exponents and diffusion coefficients, as well as the level of non-Gaussian displacements at various pH levels. Exploring such trajectories within damaged lung tissues, which exhibit compromised resistance to external pathogens, dust particles, and diesel smog, offers significant insights. This investigation could shed light on how the penetration properties of exogenous particles change within damaged lung tissue compared to healthy cells in a normal lung.

In contrast to hyperelastic models, the viscoelastic models of lung tissue need to consider the dynamic material properties, including time lag, stress relaxation, and creep, making the modeling process more complicated but more accurate [54,55]. For instance, Monjezi et al. developed an alveolar model with a three-dimensional honeycomb-like configuration. The alveolar wall was represented by linear isotropic elastic, hyperelastic, or viscoelastic material properties. Their findings revealed that lung hysteresis—the phenomenon where the lung volume during deflation is consistently greater than during inflation at the same pressure—can only be accurately captured when the alveolar wall is modeled as a viscoelastic material. This unequivocally underscores the viscoelastic nature of lung tissue’s mechanical properties and the value of viscoelastic modeling [56].

**Figure 3 bioengineering-11-00747-f003:**
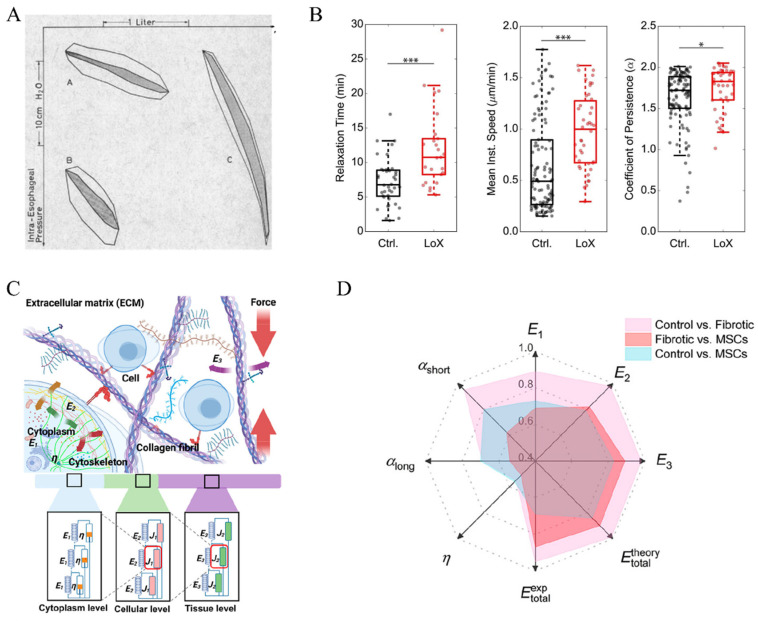
(**A**) Intraesophageal pressure–volume loops of a healthy adult (A), healthy child (B), and patient with pulmonary fibrosis (C). The work against vicious lung tissue resistance was highlighted with shaded areas [38]; (**B**) changes in the relaxation time, migration speed of cells, and cell migration persistence coefficient of a collagen-based substrate with and without treatment with LOX [44], * *p* < 0.05, *** *p* < 0.001 for Welch’s *t*-test; (**C**) the scheme of decomposing the viscoelastic properties of lung tissue via a self-similar model, with E representing elasticity, J as a composite (viscoelastic) building block, and η as cytoplasm viscosity, respectively; (**D**) an ROC analysis of how the power-law exponents (α_short_ and α_long_ for exponents within 0.01–1 s and 1–10 s, respectively) and the elasticity at the cytoplasm (E_1_), cellular (E_2_), and tissue (E_3_) levels, as well as the cytoplasm viscosity are effective in distinguishing the lung tissues of different conditions (healthy control, fibrotic, or MSC-transplanted after fibrosis). A higher value indicates greater effectiveness in distinguishing the lung tissues [49]. Reprinted and adapted from references [38,44,49] with permissions.

### 2.3. Surface Tension in Fibrotic Lungs

By 24 weeks of gestation, the inner surface of human alveoli becomes covered by the liquid surfactant produced by alveolar type-2 cells. The surfactant plays an important role in modulating the surface tension at the alveolar surface, where air and fluid meet. Without it, the air and water molecules tend to repel each other, resulting in high surface tension that causes alveoli to collapse and leads to breathing difficulty [57].

The reduction in lung surfactant is also implicated in the progression of pulmonary fibrosis. Horiuchi et al. induced pulmonary fibrosis in rats using bleomycin and found that, compared to healthy controls, the surface tension of bronchoalveolar lavage fluid (BALF) remained higher both in in vivo and in vitro experiments [58]. Horiuchi et al. induced pulmonary fibrosis in rats using bleomycin and found that, compared to the healthy control group, the surface tension of bronchoalveolar lavage fluid (BALF) remained elevated in both in vivo and in vitro experiments. The researchers perfused air and saline into the lungs and measured the pressure–volume curves of the tissue to characterize the in vivo surface tension of the lungs. The results showed that the surface tension increased significantly from 1.8 ± 0.2 to 4.7 ± 1.1 dyne/cm as the pulmonary fibrosis progressed. Additionally, the surface tension of alveolar lavage fluid was measured in vitro using a Wilhelmy balance, and the results indicated a significant increase in the mean surface tension in the bleomycin-treated group (20.7 ± 0.6 dyne/cm) compared to the control group (13.6 ± 3.8 dyne/cm). This change was accompanied by a change in the composition of the BALF. With an increase in the total protein concentration, there was a significant reduction in the concentration of phospholipids (accounting for 90% of the pulmonary surfactant and contributing to surface tension reduction) at the early stage of pulmonary fibrosis induction, accompanied by evident changes in the compositional profile [59,60,61]. Meanwhile, the concentration of surfactant protein A (SP-A) in BALF was lower in patients with pulmonary fibrosis [61]. In animal models, it was revealed that the overexpression of transforming growth factor β1 (TGF-β1), a pivotal regulator of pulmonary fibrosis, would result in the suppression of SP-A, SP-B, and SP-C expression, thereby impairing pulmonary surfactant synthesis and leading to elevated surface tension [62,63].

Notably, the inhibition of pulmonary surfactant production would also trigger pulmonary fibrosis. Li et al. investigated the role of polyhexamethylene guanidine (PHMG) in the induction of pulmonary fibrosis and found that mice exposed to PHMG aerosols for 8 weeks showed significantly impaired synthesis of SP-B and SP-C, with their binding sites blocked. These activities led to increased surface tension in the alveoli and the progression of pulmonary fibrosis [17]. They also investigated the potential role of carbon nanotube inhalation in the induction of pulmonary fibrosis. Their finding indicated that the presence of carbon nanotubes could damage the lung surfactant and induce the autophagy dysfunction of alveolar type-2 cells, which in turn could exacerbate surface tension and thereby promote the progression of pulmonary fibrosis [64]. To address this, the administration of replacement surfactants formulated by natural components has been shown to help reduce the alveolar surface tension and restore normal breathing [65,66].

The change in surface tension can also be evaluated via simulation. Based on a fluid–structure interaction analysis, Chen et al. investigated the effect of surface tension on alveolar sac mechanics. Their findings indicated that, compared to healthy lung tissue, the changes in the flow rate, volume, pressure–volume loop, and airway resistance of fibrotic lung were more affected by the changes in the tissue mechanical properties than by surface tension [67]. However, the surface tension exerted a discernible influence on the hysteretic behavior of the lung. In another study, Francis et al. investigated the properties of airflow and the air-alveolar mechanics in the presence and absence of surfactants using a computational fluid dynamics (CFD) simulation. Compared to healthy controls, insufficient surfactant resulted in a higher inflow velocity, greater vorticity, and significantly greater shear stress at the alveolar wall, thereby making the airflow more turbulent and exacerbating alveoli damage [57].

## 3. Cells Respond to Alterations in Mechanical Properties to Exacerbate Pulmonary Fibrosis

Several cell types within the lung tissue are involved in the pathogenesis of pulmonary fibrosis [68]. Typically, the process begins with the disruption of the alveolar barrier due to various triggers, followed by damage to alveolar epithelial cells. These cells interact with interstitial cells and immune cells, leading to inflammatory activities. Consequently, pulmonary fibroblasts become activated, resulting in excessive ECM synthesis, interstitial fibrosis, alveolar collapse, and the loss of pulmonary function [69]. As discussed in the previous section, an excess of the ECM alters the tissue mechanics, which can be transmitted to the cell nucleus via the cellular cytoskeleton. This, in turn, affects gene transcription and protein translation, thereby further contributing to the pathogenesis of pulmonary fibrosis [70].

This section will discuss the responses of alveolar epithelial cells, macrophages, and pulmonary fibroblasts, which are key contributors to the pathogenesis of pulmonary fibrosis, to the mechanical microenvironment in lung tissue, and to the underlying mechanisms.

### 3.1. Alveolar Epithelial Cells

Alveolar epithelial cells play a crucial role in lung growth and repair. These cells protect the lungs from the insults of the external environment by preventing the entry of foreign substances and regulating water and ion transport [71]. The alveolar epithelial cells are classified into two types: alveolar type-1 (AT1) cells, primarily involved in facilitating gas exchange, and alveolar type-2 (AT2) cells, pivotal in reducing surface tension within the alveoli by secreting pulmonary surfactants. In addition, AT2 cells can act as progenitor cells, proliferating and differentiating into AT1 cells. Normally, AT2 cells exhibit minimal self-renewal activity, maintaining a constant ratio that provides them with inherent stemness to maintain the integrity of the alveolar barrier under pathological conditions [72,73]. Since the early 20th century, researchers have gradually recognized that the primary pathogenic mechanism of pulmonary fibrosis is repeated mild injury to alveolar epithelial cells, precipitating fibrotic remodeling [74]. Watanabe et al. demonstrated that the application of a small molecule (ISRIB) accelerated the differentiation of AT2 to AT1 cells, which boosted the repair of epithelia and ameliorated the progression of pulmonary fibrosis. This finding indicates that the impediment of AT2-to-AT1 conversion may be a key event in the development of pulmonary fibrosis [75]. Wu et al. employed genetic methods to impair the regenerative function of mouse AT2 cells and observed a fibrotic progression pattern in mouse lung tissue that was similar to that of pulmonary fibrosis patients. Mechanistic studies indicated that during the process of pulmonary fibrosis, AT2 cells are exposed to a continuously elevated tension environment at the lung periphery, triggering the activation of the TGF-β pathway and thereby driving fibrosis progression from the lung periphery toward the center [29]. The finding reveals the manner in which the behavior of alveolar epithelial cells is disrupted under mechanical stress, which largely determines the pathological process of pulmonary fibrosis (Figure 4). Under mechanical stress, alveolar epithelial cells experience disruption, thereby significantly influencing the pathological process of pulmonary fibrosis. Specifically, patients with acute respiratory distress syndrome (ARDS) receiving mechanical ventilation are treated as susceptible to undergoing epithelial–mesenchymal transition (EMT), a process implicated in the development of pulmonary fibrosis [76]. Here, we focus on the impact of mechanical stimuli on alveolar epithelial cells in pulmonary fibrosis.

The lungs, functioning as gas exchange organs, experience continuous cyclic stretching throughout the respiratory cycle [77]. TMEM63 (A/B), a mechanosensitive channel protein, has been linked to the cellular response to mechanical stretching. Chen et al. found that knocking out TMEM63 (A/B) in mice resulted in atelectasis and respiratory failure. Upon mechanical stretching, the TMEM63 (A/B) channels in AT2 cells became activated, promoting the secretion of surfactants crucial for respiratory function maintenance. This study demonstrates the crucial role of appropriate mechanical stretching signals in supporting alveolar epithelial cell function [78]. Nevertheless, the progression of pulmonary fibrosis often involves excessive mechanical stretching, potentially causing oxidative damage and endoplasmic reticulum stress in alveolar epithelial cells. This condition leads to the release of damage-associated molecular patterns (DAMPs), recruiting numerous inflammatory cells and initiating the fibrotic response [79,80]. HMGB1, an important DAMP, stimulates stem cell migration and proliferation, thereby aiding in tissue repair. However, in the context of abnormal tissue repair in pulmonary fibrosis, excessive HMGB1 activates fibroblasts and enhances endothelial cell proliferation, thereby exacerbating pulmonary fibrosis [81,82]. Additionally, YAP and TAZ act as classical mechanosensors in cells, undergoing dephosphorylation in the cytoplasm when exposed to external mechanical stimuli. Subsequently, these molecules relocate from the cytoplasm to the nucleus to initiate the downstream activities [83]. For instance, the translocation of YAP/TAZ may modulate the NF-κB pathway-mediated inflammatory responses and promote the regeneration and differentiation of AT2 cells, thereby upregulating the formation of new alveolar barriers and decelerating fibrosis progression [84]. Liu et al. demonstrated that MAPK-mediated YAP translocation to the nucleus is pivotal for the upregulation of alveolar regeneration in response to lung tension [85]. Additionally, YAP can activate the mTOR/PI3K/AKT signaling pathway, promoting abnormal proliferation, migration, and EMT in alveolar epithelial cells and consequently exacerbating the progression of pulmonary fibrosis [86]. These studies indicate that mechanical stress can activate mechanosensory pathways in alveolar epithelial cells, which significantly contributes to the development of pulmonary fibrosis.

### 3.2. Fibroblast

Fibroblasts, initially observed by Virchow in the 1800s, were first characterized as spindle-shaped cells residing within tissue. The term “fibroblast” was later introduced by Ziegler to describe the cells that accumulate in the newly formed connective tissue at sites of injury [87,88]. As the most common cells within the body’s connective tissue, fibroblasts are responsible for synthesizing and secreting the extracellular matrix, thereby maintaining structural integrity [89]. Apart from this fundamental role, fibroblasts demonstrate diverse and dynamic functions, actively participating in the body’s development and intricate processes of tissue regeneration [88,90,91] (see Figure 5A).

Lung tissue possesses a certain capacity for spontaneous repair; however, pulmonary fibrosis generally involves prolonged chronic injury, with fibroblasts playing a crucial role in this process. Previous studies utilizing a model of bleomycin-induced pulmonary fibrosis in mice have identified various fibroblast subpopulations within the lungs, including ACTA2^+^ myofibroblasts, lipofibroblasts, and fibroblasts expressing COL3A1 and COL4A1 [92,93]. These studies reveal the complexity of fibroblasts in the progression of pulmonary fibrosis (see Figure 5B).

For patients diagnosed with acute lung injury and subsequent pulmonary fibrosis, the respiratory function is typically maintained via mechanical ventilation. However, it is recognized that the use of high tidal volume during mechanical ventilation may exacerbate fibrosis by significantly promoting the proliferation of fibroblasts [94]. Tang et al. conducted an in vitro study where fibroblasts were subjected to cyclic stretching at a strain of 20% and a frequency of 1 Hz to mimic the mechanical conditions experienced during high-tidal-volume mechanical ventilation. Their results revealed that excessive mechanical stimulation promotes pulmonary fibrosis through the ASK1-endoplasmic reticulum stress pathway, with inhibition of this pathway showing promising results in impeding fibrosis development [95]. Additionally, the inflammasome pathway in fibroblasts is significantly activated under excessive mechanical stimulation, leading to upregulated expression of smooth muscle actin and vimentin, further exacerbating fibrosis [96]. These results indicate that similar to AT2 cells, the changes in mechanical properties post-pulmonary fibrosis may activate fibroblasts, profoundly affecting the progression of the disease.

In patients with pulmonary fibrosis or in bleomycin-induced mouse models, Hippo-mediated mechanotransduction signaling has been reported to trigger YAP/TAZ activation, stimulating lung fibroblasts to secrete excessive ECM and leading to tissue stiffening. Furthermore, the activation of the Gα_s_-coupled dopamine receptor D1 has been shown to indirectly inhibit the function of YAP/TAZ in fibroblasts, thereby ameliorating pulmonary fibrosis [97]. Similarly, the activation of the YAP/TAZ mechanotransduction pathway has been found to promote fibrosis in other organs. NUAK family kinase 1 has been recognized as a pro-fibrotic kinase, which, in conjunction with TGF-β1 and YAP/TAZ, establishes a positive feedback loop that facilitates fibrosis progression [98]. Ryu et al. cultured lung fibroblasts from healthy donors on rigid hydrogels containing TGF-β1 to mimic the microenvironment of pulmonary fibrosis. They observed a significant elevation in the expression of the extracellular acidification rate (ECAR), ECAR/OCR ratio, and mtRNA content with increasing stiffness, indicating notable metabolic alterations. Consequently, the fibroblasts transformed into a phenotype associated with pulmonary fibrosis. A plasma sample analysis from pulmonary fibrosis cohorts revealed a correlation among stiffness, fibroblast activation, and metabolic reprogramming [99]. Li et al. studied the molecular mechanisms underlying the activation of pulmonary fibrosis via mechanical stimuli. They found that excessive matrix stiffness upregulates RhoA activity, promoting F-actin binding to the cytoskeleton and inducing the nuclear translocation of YAP via CD44, independent of the canonical Hippo pathway kinases MST1 and LATS1. These findings suggest the potential of the CD44-RhoA-YAP mechanical signaling axis as a novel therapeutic target for pulmonary fibrosis [100].

### 3.3. Macrophage

Macrophages are the most plastic cells in the hematopoietic system, playing a crucial role in growth, development, tissue repair, and immunity [101]. In normal circumstances, bone marrow monocytes migrate to various tissues, differentiating into tissue-resident macrophages, including dormant (M0) macrophages [102]. During the process of pulmonary fibrosis, alveolar epithelial damage leads to the secretion of many cytokines, driving M0 macrophages to differentiate into classically activated macrophages (M1). Subsequently, M1 macrophages produce tumor necrosis factor-alpha (TNF-a), interleukin-1 beta (IL-1β), and reactive oxygen species (ROS) to combat infection, clear foreign substances, and ultimately repair the damage [103]. Alternatively, M0 can differentiate into activated M2 macrophages, which secrete TGF-β1, platelet-derived growth factor (PDGF), interleukin-10 (IL-10), and other cytokines, promoting fibroblast proliferation and collagen synthesis for anti-inflammatory responses and tissue repair [103]. However, excessive M2 activation may result in the onset of various fibrotic diseases. Furthermore, macrophages secrete chemokine ligands 2 and 18 (CCL2 and CCL18) to recruit more monocytes from the circulation system, thereby exacerbating fibrosis [104]. Therefore, maintaining a balanced state of macrophages within the body is crucial, as any dysregulation can potentially lead to the occurrence or exacerbation of pulmonary fibrosis.

In recent years, researchers have shown increasing interest in the impact of mechanical forces on macrophages. It has been noted that M2 macrophages not only upregulate the proliferation of fibroblasts but also play a role in promoting their differentiation into myofibroblasts. The myofibroblasts are known to secrete substantial amounts of collagen, which in turn facilitates ECM deposition and remodeling by regulating the balance of matrix metalloproteinases (MMPs) [105]. Conversely, the excessive secretion and cross-linking of the ECM lead to increased matrix stiffness, thereby exacerbating the mechanosensitivity of macrophages [106]. Consequently, there is a mutual reinforcement between matrix stiffness and macrophages through positive feedback mechanisms, ultimately contributing to the development of fibrosis.

Macrophages, a group of innate immune cells that adhere to the ECM, execute their biological functions in response to various cues presented in the ECM, such as biological, biochemical, or mechanical ones (Figure 6). An increase in ECM stiffness leads to the upregulation of YAP expression and its translocation to the nucleus, triggering inflammation signals and resulting in the secretion of numerous inflammatory cytokines [107]. Studies show that ECM stiffness can influence macrophage polarization toward the M2 phenotype. Chen et al. cultured murine bone marrow-derived macrophages (BMDMs) on hydrogels with different stiffness. On soft substrates with low stiffness (2.55 kPa), macrophages showed enhanced CD86 expression, increased ROS production, activation of the NF-κB pathway, and polarization toward the M1 phenotype, leading to the secretion of IL-1β and TNF-α. Conversely, on hydrogels with intermediate stiffness (34.8 kPa), macrophages exhibited upregulated expression of CD206, reduced ROS production, polarization toward the M2 phenotype, and secretion of IL-4 and TGF-β. Furthermore, the subcutaneous delivery of two hydrogels led to the enrichment of M1 macrophages around softer hydrogels and M2 macrophages around stiffer hydrogels [108], with matrix stiffness affecting intercellular communication and modulating macrophage function. The single-cell sequencing by Taufalele et al. revealed that tumor cells in a stiffer tumor microenvironment recruit more macrophages and promote their polarization toward the M2 phenotype, suggesting that the ECM enhances this process [109]. Wang et al. showed that mechanical ventilation with a high tidal volume in mice induced pulmonary fibrosis and inflammatory responses, significantly increasing M2 macrophages in BALF, elevating TGF-β and p-Smad2/3 expression in alveolar epithelial cells, and exacerbating pulmonary fibrosis [110].

The reviewed studies demonstrate that mechanical stimuli affect various types of cells via certain mechanisms, subsequently resulting in pulmonary fibrosis. Consequently, an increasing number of studies are focusing on targeting the mechanisms by which mechanical stimuli take effect to intervene in the progression of the disease. One of the potential therapeutic targets is lysyl oxidase (LOX). A fully humanized monoclonal antibody targeting LOXL2 was developed by Gilead Sciences and named Simtuzumab. A randomized, double-blind Phase 2 clinical trial in idiopathic pulmonary fibrosis was conducted [112]. However, the progression-free survival of patients was not significantly improved with the administration of Simtuzumab. This may be due to the limited targeting sites of the LOXL2 in human studies. Therefore, more specific monoclonal antibodies need to be developed in the future. Nevertheless, researchers remain undaunted by the current outcomes and have initiated further research into LOXL2-targeting medications worldwide [113,114]. 

It is known that mechanical stimuli mediate the secretion of TGF-β by regulating integrins, which subsequently activate fibroblasts and stimulate the progression of pulmonary fibrosis [115]. Indalo developed a small-molecule drug targeting αvβ1/αvβ3/αvβ6 integrins, which showed positive results in Phase 1 clinical trials and has now entered a dosing trial for pulmonary fibrosis (ClinicalTrials.gov Identifier: NCT03949530, https://clinicaltrials.gov/study/NCT03949530, start date on 16 April 2019). Pliant’s αvβ6/αvβ1 integrin dual-selective small-molecule inhibitor, PLN-74809, is currently undergoing a Phase IIa trial among patients with idiopathic pulmonary fibrosis (ClinicalTrials.gov Identifier: NCT04072315, https://clinicaltrials.gov/study/NCT04072315, start date on 13 February 2020). The Yap/Taz signaling pathway also plays a critical role in pulmonary fibrosis. The small-molecule inhibitor of Yap/Taz, Verteporfin, has been demonstrated to reverse the progression of pulmonary fibrosis in mouse models [100]. Research on the mechanical stimulation-induced progression of pulmonary fibrosis has identified specific targets and potential pathways for drug development. With the advancement of future studies, it is anticipated that a more comprehensive understanding of various mechanisms underlying pulmonary fibrosis will be established. Along with intensive clinical research, it is expected to see significant advancement in the development of drugs for the disease.

## 4. Future Perspective

While significant progress has been made in elucidating the relationship between pulmonary fibrosis and mechanical stimuli in lung tissue, the mechanical characteristics of fibrotic lungs remain incompletely understood. Fibrosis is typically an irreversible process that progressively worsens over time. It is therefore logical to postulate that the mechanical properties of fibrotic tissues and organs evolve temporally. Although well characterized in the literature, the liver stiffness in patients increased significantly as fibrosis was exacerbated, showing a time-dependent profile [116,117]. Consequently, there is a pressing need to delineate the mechanical properties of fibrotic lungs at different stages of disease progression. Such characterization would be invaluable for studies focusing on specific phases of the disease, enabling more targeted research and potentially informing novel therapeutic strategies.

It shall also be noted that the current research primarily focuses on inducing pulmonary fibrosis in animals. While this method is advantageous for replicating the complexity of biological systems, it is also costly and resource-intensive in research facilities. There is an urgent need to develop in vitro models that are cost-effective, widely assessable, reproducible, and, more importantly, reflective of the compositional and mechanical characteristics of fibrotic lungs.

The mechanical properties of fibrotic lungs significantly differ from those of healthy lungs, characterized by increased matrix elasticity, reduced viscosity, and increased surface tension. These alterations have been studied in two-dimensional cell culture models previously (Figure 7A,B). For instance, cells were cultured on substrates with different stiffness to provide a “passive” form of mechanical stimuli (e.g., using glass as stiffer than polydimethylsiloxane) [99] or subjected to externally applied programmed mechanical loads [29]. However, the two-dimensional cell culture fails to recapitulate the actual tissue environment, where cells are embedded in a three-dimensional matrix, and mechanical stimuli are transferred in all directions rather than from a planar substrate.

To address this end, the development of a three-dimensional system with tailorable, tissue-like mechanical properties is essential, with hydrogels being one of the most commonly utilized approaches. Hydrogels consist of a network of cross-linked polymer chains derived from natural sources or synthesized. Like the natural ECM in the human body, hydrogels can effectively replicate the three-dimensional cellular environment, allowing for the modulation of mechanical properties by adjusting the degree of cross-linking in polymer chains [118,119]. For instance, Giménez et al. produced normal-like and pre-fibrotic-like 3D hydrogels by manipulating the interaction of hydrogels and encapsulated fibroblasts from patients with idiopathic pulmonary fibrosis, either allowing them to float freely or attaching them to the container. Synergistically, the presence of TGF-β1 and a pre-fibrotic-like stiff matrix subjected to mechanical resistance from the container resulted in an increased expression of COL1A1 and a reduced expression of MMP-1, demonstrating the effectiveness of a hydrogel-based system in mimicking the heightened stiffness in fibrotic lungs for mechanistic studies [120].

The development of hydrogels with time-evolving mechanical properties has made it possible to better recapitulate the mechanical stimuli as pulmonary fibrosis progresses. Enzymatic strategies can be employed to construct hydrogels with viscoelastic properties that evolve. Cacopardo et al. pioneered an approach wherein gelatin was initially cross-linked with microbial transglutaminase (mTG), then mixed with cells, and subsequently photopolymerized using ultraviolet light to produce cell-laden mTG hydrogels. These hydrogels were then immersed in culture media enriched with mTG. As time progressed, the gels exhibited increasing stiffness and viscosity, concomitant with a deceleration in cell proliferation. These findings suggest that the strategy of continuous enzymatic cross-linking through the exogenous addition of mTG effectively recapitulates the mechanical property alterations observed during disease progression in vivo [121] (Figure 7C). Another promising material is silk fibroin, which exhibits a unique property of increasing stiffness over time. This characteristic is attributed to the progressive formation of hydrogen bonds between β-sheets within the silk fibroin fibrils. This time-dependent stiffening phenomenon makes silk fibroin an excellent candidate for mimicking the evolving mechanical properties of developing tumors [122].

In addition to time-dependent properties, the spatial difference in fibrotic tissue (e.g., being stiffer in the peripheral region than the inner region and a distinctive cellular population), which is not yet satisfied with conventional two-dimensional (planar) systems or mold-cast hydrogels, may also be recapitulated in vitro. Three-dimensional bioprinting, whereby biomaterials, living cells, and biochemical factors are combined into bioinks and precisely deposited to fabricate a viable construct in a layer-by-layer manner, has demonstrated great potential in creating complex and refined structures that closely resemble natural tissues in both form and function [123] (Figure 7D). With improvements in bioink configuration and 3D printing technologies, it is now feasible to construct functional models mimetic to tissues and even organs [124]. Grigoryan et al. utilized 3D printing technology to prepare a biomimetic alveolus containing vascular and tracheal structures in vitro. A bioink based on polyethylene glycol diacrylate and food dye (as a photo-initiator) was prepared and subsequently used in projection lithography, which allows for high-precision fabrication of the alveolar structures containing extensive vascular networks. After photopolymerization, the structure was transferred into a culture medium to support the cell growth. By perfusing blood through the vessels and air into the air sacs, this biomimetic alveolus allows for oxygen delivery to the surrounding region, mimicking the respiratory process of an actual lung tissue [125] (Figure 7E). Using piezoelectric inkjet 3D printing, Kang et al. separately prepared bioinks containing alveolar cells, fibroblasts, and endothelial cells and printed them layer by layer on demand. The result is a tri-layered alveolar barrier model with a wall thickness of only 10 μm [126]. In another work, digital light processing (DLP) was employed to fabricate lung tissue-like constructs containing both blood vessels and airways. With red blood cells injected into the printed blood vessels and artificial airways ventilated, the system successfully replicated the human respiratory cycle of mixing red blood cells with oxygen and the bidirectional flow [124]. These studies demonstrate the potential of 3D printing technology to configure complex spatial features in biofabricated constructs, enabling the creation of accurate in vitro replicas of fibrotic lung structures.

**Figure 7 bioengineering-11-00747-f007:**
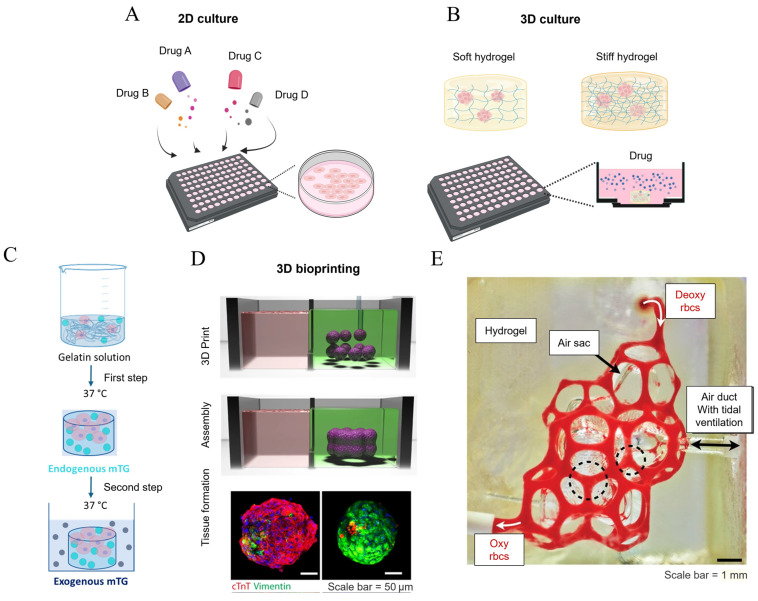
The strategies employed in constructing in vitro models of fibrotic lungs. (**A**) Conventional two-dimensional cell cultures for high-throughput drug screening. No mechanical stimulus is applied to the cells; (**B**) hydrogel-based three-dimensional constructs for cell cultures, where the mechanical properties of the hydrogel can be modulated by adjusting the hydrogel concentration, and drug screening is performed in the presence of mechanical stimuli in this setup; (**C**) a collagen-based hydrogel system with time-evolving stiffness endowed with mTG cross-linking [121]; (**D**) 3D bioprinting technology employed in the fabrication of a viable construct that recapitulates the healthy and fibrotic cardiac tissue (characterized by significant vimentin expression) [123]; and (**E**) a biomimetic, bioprinted hydrogel alveolar model. Deoxygenated blood was injected from the upper entry, and became oxygenated after acquiring oxygen from the air sacs, with the oxygen being supplied from the air duct via tidal ventilation [125]. Reprinted and adapted from references [121,123,125] with permissions.

## 5. Conclusions

It is well established that the mechanical properties of the lung undergo significant changes as pulmonary fibrosis progresses. These changes affect the elasticity, viscoelasticity, and surface tension of lung tissue under pathological conditions, significantly impacting lung cells and the fibrosis process.

As evidenced by numerous publications, alterations in the mechanical properties of the lungs can also exacerbate and promote fibrosis progression. These changes act as mechanical stimuli transmitted to alveolar epithelial cells, activated fibroblasts, and immune cells, subsequently influencing the biological activities of the cells. The investigation of the underlying mechanisms has been demonstrated to be valuable for developing novel therapeutic strategies for pulmonary fibrosis. Nevertheless, recent clinical trial results suggest that targeting a single gene or protein is not yet a sufficient therapeutic approach, necessitating further research. To this end, the scRNA-seq and spatial transcriptomic technologies are invaluable to constructing a comprehensive mechanical cellular communication atlas at multiple levels (i.e., tissue, cellular, and molecular levels in a top-down manner). The application of these technologies will allow us to decipher the pathogenic mechanisms of mechanical stimulation during the progression of fibrosis.

The progression of pulmonary fibrosis is complex and diverse, gradually advancing from an initial inflammatory stage to the ultimate fibrosis. Therefore, the mechanical properties of pulmonary tissue and, correspondingly, the mechanical cues applied to the cells shall be further investigated to trace the temporal evolution.

The quantitative investigation of the mechanical properties in fibrotic lungs is predominantly carried out using animal studies, while numerical simulation has become increasingly valuable for assessing properties and forces that cannot be directly measured at a microscopic scale. Future research shall focus on establishing numerical models of fibrotic lungs with even higher precision, a larger scale, and higher computational efficiency. It is also expected that the characteristics of pulmonary fibrosis at different stages will be recapitulated in a series of models. This will allow for the establishment of targeted research models based on the tissue mechanical characteristics of different stages and, potentially, different patients.

The interdisciplinary research that integrates cellular biology and mechanics requires the development of novel experimental models enabling full control over mechanical properties. This approach serves as an alternative to costly, facility-intensive animal studies that exhibit significant variability among individuals. The advancement of hydrogel synthesis offers new opportunities for the development of tissue- or organ-like constructs, with numerous hydrogel-based systems (e.g., Matrigel, polyacrylamide, agarose, etc.) successfully employed to simulate the tissue mechanical properties of fibrotic lungs. However, there is an imminent need for the in vitro system to take the spatiotemporal complexity of fibrotic lung tissues into account. To this end, novel hydrogels mimicking the time-evolving mechanical properties of lungs have been reported more recently, offering novel opportunities to systematically follow the cellular responses. On the other hand, 3D bioprinting technology demonstrates the capability to construct complex 3D models of lung tissue, which is highly promising to fabricate viable constructs with distinctive mechanical properties and cell populations. The fusion of time-evolving hydrogel and on-demand spatial configuration will potentially lead to a better representation of the fibrotic lung environment, thus providing a new way to investigate the interactions between cells and mechanical forces.

In conclusion, the complex mechanical interactions inherent in lung fibrosis demand intensive investigation in the future. The utilization of controllable mechanical stimuli upon innovative in vitro models remains a valuable approach for understanding the impact of altered mechanical properties within the lung on cells, disease progression, and critical biological targets, thereby informing corresponding therapeutic strategies.

## Figures and Tables

**Figure 2 bioengineering-11-00747-f002:**
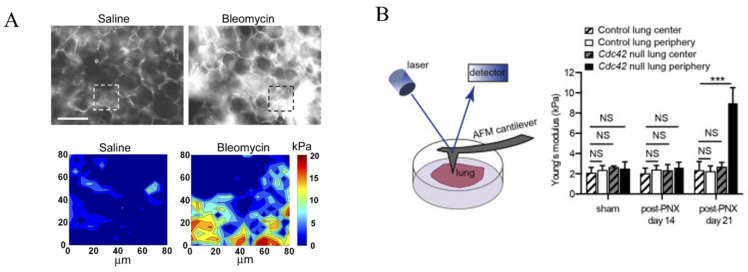
(**A**) Immunostaining of collagen I in murine lung parenchyma treated with either saline (left) or bleomycin (right, for pulmonary fibrosis induction), as well as the corresponding AFM elastograph of the region of interest as highlighted with dashed squares. Scale = 100 μm [28]. (**B**) The scheme of investigating the Young’s modulus of lung using an AFM, and the Young’s modulus of lungs of different regions in control and Cdc42-null mice following sham surgery or post-lung lobe resection (PNX) [29]. Reprinted and adapted from references [28,29] with permissions. NS: *p* > 0.05, *** *p* < 0.001 for Student’s t test.

**Figure 4 bioengineering-11-00747-f004:**
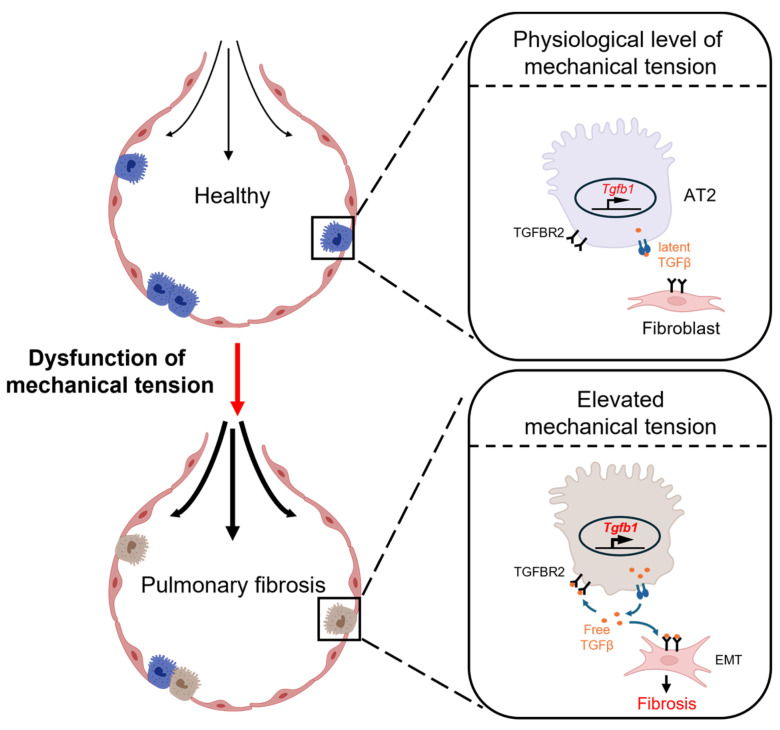
A schematic view of how elevated tension in fibrotic lung tissue exacerbates pulmonary fibrosis [29]. The elevated tension in fibrotic lung tissue activates AT2 cells in the peripheral lung tissue to secrete TGF-β, which subsequently activates the fibroblast to exacerbate pulmonary fibrosis.

**Figure 5 bioengineering-11-00747-f005:**
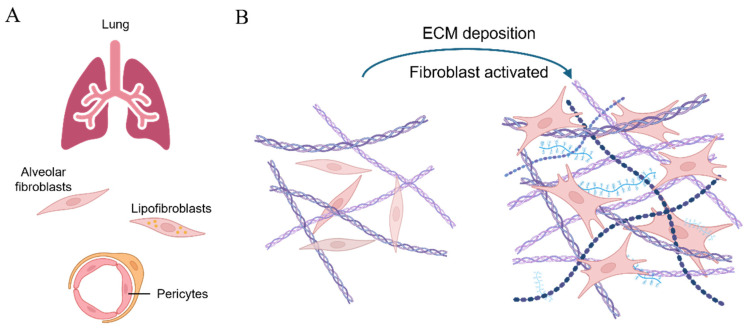
A scheme of cells differentiating into fibroblasts and participating in fibrosis progression [88]. (**A**) Different types of mesenchymal cells in lung tissue, including alveolar fibroblasts, lipofibroblasts, and pericytes, can be activated and transformed into myofibroblasts upon stimulus; (**B**) the extensive deposition of the ECM activates the transformation of fibroblasts into myofibroblasts in lung fibrosis, thereby exacerbating the condition.

**Figure 6 bioengineering-11-00747-f006:**
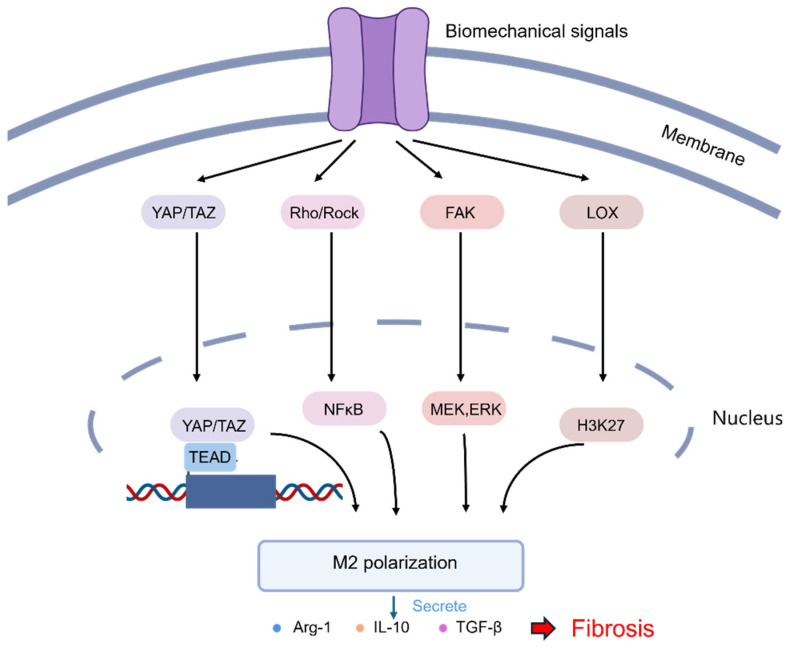
A scheme illustrating the major pathways in which mechanical stimuli regulate the polarization of macrophages through mechanoreceptors [111]. The activation of mechanoreceptors in macrophages transduces mechanical signals to the nucleus through different downstream response proteins, mediating the polarization of macrophages into the M2 phenotype.

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
