# Peer review of "Biomechanical Properties and Cellular Responses in Pulmonary Fibrosis"

_bioengineering, 2024, doi:10.3390/bioengineering11080747_

Round 1
Reviewer 1 Report
Comments and Suggestions for Authors
This work is precious for progression to combat lung fibrosis diseases.
Reviewer 2 Report
Comments and Suggestions for Authors
The paper is generally well presented.
in section 2, I would include variation in the viscosity of the liquid phase of the ecm and of the crosslinking degree of elastic fibers related to lox upregulation (not only the increase of the number of fibers).
I suggest to deepen the analysis on viscoelastic testing, see for instance https://www.mdpi.com/2076-3417/14/3/1093
I also suggest to include in the last section related to in-vitro design the possibility of replicating the evolution of mechanical properties of fibrotic tissues over times thanks to enzymatic stiffening strategies (see for example https://www.mdpi.com/2306-5354/8/8/106)
Reviewer 3 Report
Comments and Suggestions for Authors ​ This review article provides a broad coverage of multiple bio-mechanical properties and of cellular responses in pulmonary-fibrosis cells. The material is publishable, but it requires a revision, please address the points below. As the initial, introductory figure the authors are encouraged to present a picture of a healthy versus a damaged lung tissue and of the respective cells as well as of their physical-chemical-mechanical-etc. properties. This will attract more readers to the considerations which follow later in the text. As a review, i found the material somewhat "watery", with not much substance: the authors should deeply think how to add more concrete facts and more quantitative information. So far, only very qualitative facts and some trends are presented, without any coverage of systematic large-scale results. For instance, regarding mucus layers and about hydrogels in general one can and should present much more quantitative information. Both the experimental and theoretical observations should be covered in the revised version. For instance, the results of rheological experiments and of single-particle-tracking measurements were studied in Ref. [​https://doi.org/10.1039/C8SM02096E ] which should be mentioned in the revised version. The analysis of such trajectories on damaged lung tissues---with a reduced resistance to external pathogens, dust particles, diesel smog, etc.---would be very desirable. This would allow to see how the properties of penetration of exogenous particles in a damaged lung tissue changes as compared to healthy cells in a healthy lung. Conclusions should be extended, including listing concrete future perspectives. The list of abbreviations is apparently not complete.
Reviewer 4 Report
Comments and Suggestions for Authors
The authors must be complimented for a very clear and up to date review about a very important issue that goes beyond the clinical relevance. As a matter of fact, mechanobiology is one of the most promising (and relatively unexplored) research avenue in biological sciences and this review can be a very useful primer for scientists interested to follow this direction.
Round 2
Reviewer 3 Report
Comments and Suggestions for Authors
This is very solid revision, in particular the introduction and the conclusions sections were improved dramatically: the current version can be accepted for publication.